# Selective Detection of Folic Acid Using 3D Polymeric Structures of 3-Carboxylic Polypyrrole

**DOI:** 10.3390/s20082315

**Published:** 2020-04-18

**Authors:** Gheorghe Melinte, Andreea Cernat, Maria-Bianca Irimes, Szabolcs János Györfi, Mihaela Tertiș, Maria Suciu, Liana Anicăi, Robert Săndulescu, Cecilia Cristea

**Affiliations:** 1Analytical Chemistry Department, Faculty of Pharmacy, Iuliu Haţieganu University of Medicine and Pharmacy, 4 Louis Pasteur St., 400349 Cluj-Napoca, Romania; Melinte.Gheorghe@umfcluj.ro (G.M.); Ilioaia.Andreea@umfcluj.ro (A.C.); irimesmaria4@gmail.com (M.-B.I.); szabolcsjanos.gyorfi@gmail.com (S.J.G.); mihaela.tertis@umfcluj.ro (M.T.); 2National Institute for Research and Development of Isotopic and Molecular Technologies, 67-103 Donat St., 400293 Cluj-Napoca, Romania; maria.suciu@itim-cj.ro; 3Biology and Geology Faculty, Babes-Bolyai University, 44 Gheorghe Bilaşcu St., 400015 Cluj-Napoca, Romania; 4Center for Surface Science and Nanotechnology, University Polytechnica of Bucharest, 313 Splaiul Independentei St., 060042 Bucharest, Romania; lanicai@itcnet.ro

**Keywords:** electrochemical sensor, polypyrrole, folic acid, pharmaceutical products

## Abstract

The detection of folic acid in biological samples or pharmaceutical products is of great importance due to its implications in the biological functions of the human body, along with the development and growth of the fetus. The deficiency of folic acid can be reversed by the intake of different pharmaceutical formulations or alimentary products fortified with this molecule. The elaboration of sensing platforms represents a continuous work in progress, a task in which the use of conductive polymers modified with different functionalities represents one of the outcoming strategies. The possibility of manipulating their morphology with the use of templates or surfactants represents another advantage. A sensing platform based on carboxylic functionalized polypyrrole was synthesized via the electrochemical approach in the presence of a polymeric surfactant on a graphite-based surface. The sensor was able to detect the folic acid from 2.5 μM to 200 μM with a calculated limited of detection of 0.8 μM. It was employed for the detection of the analyte from commercial human serum and pharmaceutical products with excellent recovery rates. The results were double checked using an optimized spectrophotometric procedure that confirmed furthermore the performances of the sensor related to real samples assessment.

## 1. Introduction

Folic acid (FA), known as pteroylglutamic acid or Vitamin M or B9, is a B type water-soluble vitamin with important functions for the human body. Beside the implications in DNA synthesis, it has a major role in the development and healthy growth of the fetus [1]. The studies describe that pregnancy complications are closely linked with deficiency of FA, and that the daily supplementary intake of FA (400 μg) decreases the risk of neural tube defects by 50%. Moreover, congenital heart disease, preeclampsia, preterm labor, and recurrent pregnancy loss were also linked to the deficiency of this compound [2]. FA cannot be stored in the human body, thus the regular intake from fortified dietary products or vitamin supplementation is necessary for a healthy lifestyle. Also, it is known that folates can be easily degraded during storage due to heat, oxygen, and light [3], thus the necessity to develop quantification methods that can be applied for biological samples, drug and food control. The most common methods for the quantification of FA are: spectrofluorimetry [4], spectrophotometry [5,6,7], high performance liquid chromatography (HPLC) tandem mass spectrometry [3], ultrahigh performance liquid chromatography with conductivity, and UV–Vis detectors [8].

The recommended method by the pharmaceutical industry for the quantification of FA is represented by HPLC with a UV-VIS detector, as stated by European, British, and US pharmacopoeia [9]. Even though the method itself is well-known as sensible and selective, it has a number of drawbacks related to the high cost of the supplies, laborious operation protocols, increased analysis time, and the need of highly trained personnel. Also, the amount of residual organic solvents and the cost associated to their disposable raises concerns regarding their use in biomedical and pharmaceutical analysis, where the quantification of FA has drawn significant attention.

Electrochemical sensors have the ability to overcome the difficulties stated above while maintaining the same analytical performances. The high selectivity of this method, associated with the low cost of the analytes and reduced amount of residues are reasonable arguments towards the development of electrochemical sensors for the detection and quantification of FA. Up until now, there are several approaches for the development of electrochemical sensors for the same analyte including associations of different metallic or carbon-based nanomaterials: NiO and graphene, lanthanum doped ZnO with graphene, Au nanoparticles, ZnO-CuO nanoplates, ZnO-carbon nanotubes, Bi layer deposited mainly on glassy carbon or carbon paste electrodes [10,11,12,13,14,15]. An alternative to the conventional electrodes that are mainly used for experimental work in research laboratories, is the use of disposable commercial or home-made screen-printed electrodes that can simplify and optimize the testing protocols, while also diminishing the analysis cost. The decontamination procedures that are mandatory when using biological samples can be eliminated in the case of single-use electrodes [16].

To our knowledge, the use of conductive polymeric structures for the detection of FA has not been reported until now. The high conductivity, biocompatibility, and the possibility to control the final configuration are remarkable advantages for electrochemical applications of these polymers [17]. The polypyrrole-based nanostructures can be generally synthesized via chemical or electrochemical methods using different kinds of templates and they can have different features depending on the final configurations [18]. The use of a polystyrene beads template for the polymerization of pyrrole generated a honeycomb-like, highly-ordered structure that was employed for the detection of paracetamol. Despite the results, the removal of the template in an organic solvent for 10–12 h represented a major drawback that increased the time of elaboration [19]. The structuration of polymeric films in the absence of a template represents an important route for the generation of different morphologies. The generation of polymeric nanoparticles via multipulse amperometry represents a controlled method to increase the active surface area of the electrode available for further functionalization or detection of established targets [20]. One of the requirements of the electrochemical structuration of the polymeric film is the presence of surfactants that orientate the growth of the polymeric layer perpendicular on the surface of the electrode. Overall, the negatively charged polar groups of the anionic surfactant orientate the perpendicular growth, while in the case of polymeric chained surfactants, the interaction of the chains with polypyrrole enables the synthesis of cone-like structures [21].

Herein, we report the development of an electrochemical sensor for the detection of FA based on polypyrrole 3D structures with carboxylic moieties generated in the presence of a polymeric surfactant that orientates the growth of the structures perpendicular on the working surface and not as a continuous film.

## 2. Materials and Methods

The reagents were purchased and used without any further purification and pretreatment steps. The reagents: 3-carboxylic pyrrole, K_4_[Fe(CN)_6_], K_3_[Fe(CN)_6_], Na_2_HPO_4_, NaH_2_PO_4_, LiClO_4_, polyvynilpyrrolidone (PVP) were purchased from Sigma Aldrich. FA was purchased from Toronto Research Chemicals Inc. and the commercial human serum from Thermo Scientific and kept at −20 °C. The pharmaceutical tablets (Acifol, 5 mg, Zentiva) were purchased from a local drug store and kept at room temperature. All the solutions were prepared using with Milli-Q ultrapure water (18 MΩ cm^−1^).

The screen-printed electrodes (Metrohm Dropsens (Spain)) have a carbon-based working surface with a geometrical surface of 12.56 mm^2^, silver pseudoreference electrode, and carbon auxiliary electrode.

The electropolymerization of 3-carboxylic pyrrole was carried on by cyclic voltammetry (CV), according to a published protocol by Wysocka-Żołopa et al. [21]. A 50-μL drop containing 0.1 M monomer in 0.1 M LiClO_4_ and 9 mg/mL PVP, as polymeric surfactant was drop casted on the working surface of the electrode and the potential was scanned ten times from −0.5 to 1.1 V/Ag, with a scan rate of 100 mV s^−1^. The electrochemical characterization of the modified electrodes was performed by CV and electrochemical impedance spectroscopy (EIS) after drop casting a 50-μL drop of 5 mM ferrocyanide/ferricyanide ([Fe(CN)_6_]^3-/4-^) in 20 mM phosphate-buffered saline (PBS) pH 7.4 solution on the working surface. The potential was then scanned from −0.4 to 0.9 V/Ag, three times with 50 mV s^−1^. The EIS experiments were performed in the presence of the same redox mediator and conditions on a frequency window of 10 mHz to 100 KHz at open circuit potential.

Before the detection of FA, a pretreatment differential pulse voltammetry (DPV) procedure in 50 μL of the solution containing the target analyte from −0.8 to 1 V/Ag with a scan rate of 20 mV s^−1^ was done. After this step, the detection of FA was achieved by CV in the same solution. The potential was scanned three times between −0.8 and 1 V/Ag starting from 0 V to 1 V/Ag, with a scan rate of 50 mV s^−1^. The sensitivity of the platform for FA detection was assessed by using standard solutions prepared in 20 mM PBS pH 7.4, as follows: 2.5 μM, 5 μM, 12.5 μM, 25 μM, 50 μM, 100 μM, 150 μM, and 200 μM.

The real samples analysis was performed on commercial human serum as well as on pharmaceutical products (tablets). The commercial human serum was diluted with 20 mM PBS at 1:100 ratio prior to the analysis and then spiked with the standard solutions of FA in order to achieve the following concentrations: 5 μM, 25 μM, 50 μM, 100 μM, and 200 μM.

The samples from pharmaceutical products with a declared concentration of 5 mg FA/tablet were prepared according to the following protocol: 10 tablets were weighed and grinded into a fine powder. An amount of 100 mL deionized water or 20 mM PBS pH 7.4 were added and the resulting suspension was ultrasonicated for 90 min and then filtered through 0.45 μm and 0.2 μm Phenex microfilters until a clear solution was obtained. In the first case, where deionized water was the extractive solvent, the solution was diluted 1:45 with 20 mM PBS pH 7.4 in order to obtain a theoretical concentration of 25 μM of FA. The sample was evaluated, according to the same protocol applied in the case of the standard solutions, immediately after the extraction, because FA solutions are known to easily degrade due to light and temperature [6,22]. After that, the samples were spiked with an internal standard of FA in order to achieve a theoretical concentration of 100 μM. The second solution obtained after the extraction in PBS was further diluted with the same solvent in order to generate solutions with theoretical concentrations of FA, as follows: 25 μM, 50 μM, and 100 μM, and then the steps described above were followed.

The stability of the platform towards the detection of FA was evaluated by performing successive tests on the same electrode using a 100 μM standard solution of FA. The stability in time was assessed after 1, 3, 10, and 30 days after the elaboration of the platform. Each experiment was performed on a new electrode.

The electrochemical experiments were carried on Autolab PGSTAT30 and Autolab PGSTAT302N potentiostats equipped with an EIS module.

The scanning electron microscopy (SEM) images for the surface characterization of the electrodes were registered on SU8230 SEM (Hitachi, Japan). Samples were mounted on SEM specimen holders and grounded with a carbon tape. For imaging, samples were sprayed with a 10-nm gold layer by sputter-coating. Before imaging, uncoated samples were spectroscopically analyzed using Oxford Instruments X-ray detector and AZtec software for elemental composition (EDX analysis). All analysis was made at 15-mm working distance and 30-kV accelerating voltage.

Spectrophotometric determinations were performed using a SPECORD250 PLUS (Analytical Jena, Germany) spectrophotometer fitted with deuterium and tungsten lamps and equipped with WinAspect software. Quartz cuvettes with optical path length of 1 cm were used.

## 3. Results and Discussion

### 3.1. Electrochemical Deposition and Characterization of the Polypyrrole Morphology

The electrochemical deposition of 3-carboxylic pyrrole was performed by CV in a 0.1 M LiClO_4_ electrolyte solution in the presence of PVP, as polymeric surfactant [21]. The potential was scanned 10 times from −0.5 V to 1.1 V vs. Ag with a scan rate of 50 mV s^−1^. As it can be seen in Figure 1a, a peak couple corresponding to the growth of the polypyrrole backbone was observed. The anodic (0.07 V/Ag) and cathodic (−0.07 V/Ag) peaks corresponding to the oxidation/reduction processes linked to the electrochemical deposition of the polymeric structures registered an increase with every scan suggesting the growth of the conductive layer on the electrode carbon surface. The anodic peak underwent a slightly anodic shift of 0.03 V (from 0.05 V (1st cycle) to 0.08 (10th cycle), while the cathodic one had a similar cathodic shift of 0.06 V (from −0.033 V (1st cycle) to −0.1 (2nd cycle)) suggesting the fact that the polymeric structures did not reduce the electron transfer rate, despite the increase of polymerization signal. This may be evidence of the conductive nature of the polymer generated at the surface of the electrode [23,24].

Several control experiments were performed in order to prove the successful polymerization of 3-carboxylic pyrrole monomer. The deposition was carried out in the absence of the surfactant and the current intensity for the 10th cycle was significantly lower compared to the deposition in the presence of PVP, as it can be observed in Figure 1b. Another control experiment was performed in the same conditions, but in the absence of the monomer in order to assess the influence of the surfactant and electrolyte. As expected, no peaks characteristic to the electrochemical polymerization at around 0 V/Ag were observed, confirming that the peak couple mentioned above corresponds only to the electrochemical generation of the polypyrrole (Figure 1b).

The concentration of the monomer (0.1 M) was established based on the SEM images and the response towards the detection of FA. As it can be seen in Figure 1c, with the increase of the concentration of the monomer it was observed an increase of the capacitive current that could block the signal attributed to the electrochemical oxidation of FA.

### 3.2. Characterization of the Platform

The platforms were assessed by electrochemical and microscopic methods. The polypyrrole structures were tested in the presence of a redox probe, 5 mM ([Fe(CN)_6_]^3-/4-^ solution in 20 mM PBS pH 7.4, to evaluate the electron transfer rate properties before and after the electrochemical generation of the polymeric film and also after the detection of FA (Figure 2a). The slight decrease after the polymer deposition, suggested that the modifications performed at the electrode surface reduced the electron transfer rate due to the different morphologies that partially covered the graphite-based working surface. After the detection of 100 μM FA (a DPV preconditioning step and CV procedure, as described in the Materials and Methods section), a supplementary increase of the anodic/cathodic peaks was observed corresponding to the oxidation/reduction of the redox probe. This change could be attributed to the polymerization or adsorption of FA on the working electrode that increased the electron transfer rate [25].

The interfacial charge-transfer phenomena was assessed step by step on the unmodified carbon-based electrode, after the deposition of the polymeric structures and after the testing of FA by EIS experiments in a 5-mM [Fe(CN)_6_]^3-/4-^ solution in 20 mM PBS pH 7.4 (Figure 2b). The parameters of the Randles equivalent circuit were evaluated using NOVA1.10.4 software by fitting the model with the experimental data and are presented in Table 1.

The circuit for the bare graphite-based electrode was determined as [Rs(Q[R_ct_W])] (Rs-the resistance of the electrolyte solution, Rct-the charge transfer resistance at the electrode/solution interface, and W-the Warburg impedance). The classical capacitance C was replaced by a constant phase element (Q) as a consequence of the high irregularity of the carbon substrate. After the deposition of the carboxylic pyrrole there were included new series elements as follows: a pair of (RQ) and one of (RC) determined by the changes in the morphology of the electrodes surface after the electropolymerization ([Rs(Q_1_[RctW])(R_1_Q_2_)(R_2_C)]). A dramatical decrease of Rct after the deposition step from 508 Ω to 55 Ω was observed and confirmed the improvement of the electron transfer rate. The second semicircle in the Nyquist plot can be connected to the generation of the polymeric 3D structures (observed in Figure 3D–I). After the FA testing, a supplementary series of pairs of elements (RQ) was included in the equivalent circuit, [Rs(Q_1_[RctW])(R_1_Q_2_)(R_2_C)(R_3_Q_2_)], determined by the overoxidation of the polypyrrole that takes place at 0.8–0.9 V or by the possible polymerization of FA, as already described in literature data [23,25]. The decrease of Rct to 11.2 Ω proved that the FA layer had a positive effect on the electron transfer rate.

Globally, the results obtained by EIS studies confirmed the same dynamic in the electron transfer rate correlated to the surface modification with the data obtained by CV.

The SEM images (Figure 3) performed at different magnifications on the platforms generated after the polymerization in the presence of PVP revealed 3D structures (200 nm) dispersed on the rough surface of the carbon-based screen-printed electrode, different than those mentioned on the literature due to the different substrate [21]. Also, the presence of the carboxylic moieties can have an influence on the morphology of the polymer modifying the final configuration.

The EDX data indicated the presence of the polymeric structures on the working surface by the presence of nitrogen. The measured weight percent of N (found only in the polymeric structures) on the unmodified electrode and the electrodes modified with 0.05 M and 0.25 M 3-carboxylic pyrrole in 9 mg/mL PVP in 0.1 M LiClO_4_ was: 0%, 8.56%, and 12.65%, respectively. The progressive increase of the monomer concentration confirms the deposition of the polymeric structures. The same trend was observed for the weight percent of O (found in the carboxylic functionalities), which highlights the deposition of a higher amount of polymer.

### 3.3. The Electrochemical Oxidation of FA

The electrochemical behavior of the analyte was studied on the modified electrodes with 3D polymeric morphologies. The first step consisted in a conditioning DPV, followed by a CV procedure. If the CV was performed prior to the conditioning step, no oxidation peak at −0.5 V/Ag was observed and therefore the DPV was considered mandatory for the detection of FA.

A well-defined anodic peak at −0.2 V/Ag was observed during the DPV analysis and was linked to the electrochemical oxidation of the polypyrrole backbone. According to studies from the literature, the electrochemical oxidation of FA to dehydrofolic acid takes place at around 0.7–0.8 V independent of the pH value [25,26,27,28], by losing two protons (H^+^ of C(9) and H^+^ of N(10)) and two electrons (Scheme 1, compound II) [22,26]. In our study, this process cannot be observed due to the capacitive current determined by the polymeric layer that blocks the signal of the anodic peak of the analyte. The DPV anodic pretreatment performed at low scan rate enabled the oxidation of FA molecules at the electrode surface. This oxidation procedure seemed to determine the activation of the redox process of FA, since in its absence the signal was substantially lower. A higher amount of dehydrofolic acid was also generated by CV when the potential was firstly scanned in the anodic domain. 

The detection of FA via CV is in agreement with the results presented in literature data, as follows: a cathodic peak at −0.17 V/Ag corresponding to the redox process of the polypyrrole backbone, at −0.37 V/Ag (Scheme 1, compound III) and −0.6 V/Ag (Scheme 1, compound IV) corresponding to the electrochemical reduction of pterin moiety of FA (the processes involve the gain of two electrons and two protons) that generates 5,8 dihydrofolic acid. As already reported, the 5, 8-dihydrofolic acid undergoes a pH dependent tautomerization process to 7,8 dihydrofolic acid. At pH 7.4, this process is diminished in comparison to an acidic pH and the further reduction of the tautomerization process does not take place [25].

The oxidation process highlighted the reversible oxidation of 5,8-dihydrofolic acid at −0.5 V/Ag. At 0.1 V/Ag, the anodic peak corresponding to the oxidation of the polymeric structures was also observed.

No anodic oxidation of the molecule was reported either on the second oxidation step of the CV due to the polymer that is prone to overoxidize at 0.7–0.8 V/Ag and hinders the electrochemical signal of the analyte [25,26,29,30]. The polymeric structures undergo an overoxidation process at 0.8 V/Ag and the capacitive current blocks the potential anodic peak of the analyte and no signal was detected.

As it can be seen in Figure 4b, the anodic peak at −0.5 V/Ag associated to the electrochemical oxidation of FA was observed only on the configuration where the electrochemical polymerization was performed in the presence of PVP, as a control experiment. The 3D morphologies have an important effect on the detection of FA, confirmed by the fact that no anodic peak was detected when the 3-carboxylic pyrrole monomer was polymerized in the electrolyte solution lacking the surfactant or on the unmodified electrode. Thus, the interaction with the polymeric structures enabled the reversible oxidation of 5, 8 dihydrofolic acid at −0.5 V/Ag. Clearly only a thin film generated in the absence of PVP is not sufficient for the detection of FA, the 3D structures have a superior ability to interact via electrostatic bonds with the analyte, maybe due to the increased density of carboxylic moieties.

The same experiment was performed in the presence of an FA solution degassed with nitrogen for five minutes. No difference between the two voltammograms performed in the same conditions, only with a degassing step in the second case was observed underlining that the reduction of oxygen did not have any influence on the redox processes of FA (Figure 4b blue vs. light blue line).

The intensity of the anodic peak correlated with the electrochemical oxidation of 5, 8-dihydrofolic acid increased proportionally with the increase of the concentration of the analyte from 2.5 μM to 200 μM (Figure 4c). The equation of the linear regression curve was determined as: I (μA) = 0.0245 * [FA] (μM) + 0.731 and the calculated limit of detection (LOD) was 0.8 μM (estimated based on the signal/noise ratio of three, S/N = 3) (Figure 4d).

FA is a molecule that is strongly prone to the adsorption on the surface of the electrode due to the p-aminobenzoic acid moiety [25] and the high irregularity of the graphite-based working electrode. The current intensity corresponding to the anodic peak at −0.5 V increased proportionally with the scan rate confirming furthermore the adsorption of the analyte on the polymeric platform (Figure 4e). When plotting the current intensity against the square root of the scan rate, no linear correlation was observed, underlining that the electron transfer is not diffusion controlled, which means that FA molecules remain adsorbed on the electrode surface (Figure 4f).

### 3.4. Intraassay Stability

The intraassay stability of the sensor was assessed by repetitive tests on the same electrode using each time a different standard solution of 100 μM FA. Between the assays, the electrode was thoroughly rinsed with ultrapure water. After the second test, it registered an increase of the signal with 40% and after the third test an increase with 52% (Figure 5a). 

The results are in agreement with the polymerization of FA on the polymeric substrate that acted as a preconcentration step, and in this case, the sensor can be used for only one determination. Despite this fact, the manipulation of biological samples involves safety issues related to the user and environment contamination and the reuse of the same sensor could raise difficulties in this direction. Thus, the home-made or commercially disposable screen-printed electrodes ensure the successful determination of FA from human serum or pharmaceutical formulations with minimum contamination risks and low elaboration costs.

### 3.5. Stability in Time

The stability in time was evaluated on different sensors prepared on the same day and tested after 1, 3, 7, 10, 30 days in the presence of 100 μM FA, while being kept at 4 °C. The results indicated an excellent recovery of the signal in comparison with the data obtained in the first day (Figure 5b), with less than 2.5% difference in each case, suggesting that after the elaboration of the platform it remains stable up to one month and it can be used without any further pretreatment, other than the two-step detection procedure.

### 3.6. Interference Studies

The selectivity of the sensor towards the detection of FA was assessed in the presence of dopamine, serotonin, and ascorbic acid, analytes commonly found in real samples such as biological fluids. The experiments were performed in the presence of 25 μM interfering agent and 100 μM FA. It can be easily observed that the anodic peak corresponding to 5,8-dihydrofolic acid oxidation had almost the same intensity in the presence of dopamine, serotonin, and ascorbic acid (Figure 5c). The oxidation peaks of the analytes were observed, in agreement with literature data, at 0.1 V/Ag for dopamine (red line), at 0.3 V for serotonin (green line), and at 0.1 V/Ag for ascorbic acid (blue line), but no interferences with the electrochemical signal of FA were observed [17,31]. Thus, the broad separation (0.6–0.8 V) between the anodic peak at −0.5 V/Ag and the ones corresponding to the studied interfering agents (0.1–0.3 V/Ag) allows their simultaneous analysis without any other modification of the platform. The average recoveries of the signal corresponding to the oxidation of FA in the presence of dopamine, serotonin, and ascorbic acid were 100.49% (RSD 4.98%), 94.52% (RSD 5.13%), and 95.12% (RSD 3.74%), respectively (Figure 5d).

It needs to be mentioned that the presence of carboxylic groups on the polymeric structures is able to electrostatically reject compounds with same functionalities (like ascorbic acid) when working at physiological pH, contributing to the same global results [32]. Other studies also highlighted the simultaneous determination of FA in the presence of ascorbic acid and serotonin taking into account only the anodic peak at 0.7–0.8 V corresponding to the oxidation of the analyte to dehydrofolic acid, without any mention of the anodic peak at −0.5 V/Ag that corresponds to the reversible oxidation of 5,8-dihydrofolic acid, as presented in the literature [11].

### 3.7. Real Sample Analysis

#### 3.7.1. Commercial Human Serum

The electrodes were used in order to assess the concentration of FA in real samples like commercial human serum and pharmaceutical products. The human serum samples were diluted 1:100 with 20 mM PBS pH 7.4 and then spiked with FA to achieve concentrations of 5 μM, 25 μM, 50 μM, 100 μM, 200 μM.

It can be seen that no anodic peak was observed at −0.5 V/Ag in the absence of the target analyte highlighting that the biological matrix does not interfere with the determination of FA (Figure 6a). The obtained results showed a very good recovery for the current intensity of the anodic peak: 101.9% (RSD 2.44%). Moreover, the current intensities of the anodic peaks were plotted in the calibration curve obtained with the FA standard solutions and the obtained results indicated an average recovery of the concentrations of 106.0% (RSD 16.7%). The obtained values were marked in red in the linear regression curve representation for the standard solutions of the analyte (see Figure 4d, red plots for 5 μM, 25 μM, 50 μM, 100 μM, 200 μM concentrations).

The dilution ratio was chosen after an optimization step where no characteristic anodic peak was obtained on the undiluted serum, while only 60% of the signal was observed when the dilution ratio was 1:10, as can be seen from Figure 6b. The diminution of the signal can be attributed to the presence of the proteins in serum that reduced the electron transfer rate. The equation of the calibration curve in serum was determined to be I (μA) = 0.024 * [FA] (μM) + 0.874 (R² = 0.994).

The matrix effect was evaluated using a strategy proposed by Kmellar et al. [33,34]. The matrix coefficient was calculated as follows: matrix effect coefficient (%) = (1 − slope matrix/slope solvent) * 100, where the slope matrix represents the slope of the calibration curve performed in commercial human serum spiked with different concentrations of FA, and the slope solvent represents the slope of the calibration curve performed in standard solutions of FA prepared in 20 mM PBS pH 7.4. The obtained result of 1.6% indicated that the matrix, in this case the commercial human serum diluted 1:100, had no effect on the analysis of FA, knowing that the matrix effect could be considered negligible from −20% to 20% according to Kmellar et al. [33].

Even though the matrix has a low influence on the quantification of FA, the reference FA level in blood is in the 0.01–0.079 μM range [35], lower than the linear range for the developed sensor (2.5–200 μM). In this case, the ability of the platform to assess FA from human serum can only be exploited in the presence of an internal standard.

#### 3.7.2. Pharmaceutical Products

FA commercial tablets were prepared for analysis using ultrapure water as the extraction solvent, as described in the Materials and Methods section and then tested and the anodic current intensity was read.

The current intensity of the anodic peak was plotted in the equation of the calibration curve (I (μA) = 0.025 * [FA] (μM) + 0.731) and the result indicated a concentration of 24.44 μM. This result reflects a very good recovery of FA in the commercial tablets of 96% (RSD 16.55%) (calculated after taking into account the dilution steps). In the sample solution was added an internal standard of FA as to obtain a theoretical total concentration of 100 μM FA. The current intensity was plotted in the equation of the calibration curve and the result indicated an average concentration of 104.65 μM FA that represents a recovery of about 104.6%.

The influence of the extraction solvent was also assessed when using 20 mM PBS pH 7.4 instead of deionized water on the same 5-mg FA tablets. After following the same filtration step, the samples were diluted as to achieve theoretical concentrations of 25 μM, 50 μM, and 100 μM. The intensity of the signal corresponding to the 5,8-dihydrofolic acid electrochemical oxidation was plotted in the calibration curve and the corresponding concentrations of the samples were calculated. The average recovery for the calculated concentrations was 102.1% (RSD 3.2%, 5 samples) and for the amount of FA in a tablet the average recovery was 100.2% (RSD 3.3%, 5 samples). The results underlined that the extraction of the FA in PBS proved to be more efficient than in deionized water, probably due to the presence of the salts that facilitated the extraction of the target molecule. The results obtained via the electrochemical method were summarized in Table 2.

The electrochemical experimental data obtained when testing pharmaceutical products containing FA, were confirmed by evaluating the same real samples by UV-VIS spectroscopy. Prior to this step, the absorbance maximum for the FA molecule was determined at 282 nm, which was selected as the detection wavelength for the following steps: calibration curve and real samples analysis. In these conditions, Beer’s law was obeyed in the concentration range from 5 to 75 μM FA with an excellent R2 of 0.996 (A = 0.031 * [FA] (μM) − 0.022). The clear solution obtained after the extraction protocol in PBS was diluted with PBS in order to obtain 25-μM and 50-μM theoretical concentrations of analyte. The average absorbance values for the above-mentioned concentrations were: 0.782 and 1.481, respectively. The corresponding concentrations were then calculated from the calibration curve and an average recovery of 101.0% (RSD 5.1%, six samples) it was obtained, and for the amount of analyte from the 5-mg commercial tablets, an average recovery of 97.7% (RSD 3.7%, 6 samples) was calculated (Table 2).

The recoveries for the FA amount in tablets obtained via the electrochemical method were compared with those obtained using the UV-VIS method. T-Test (two-sample assuming equal variances) was performed and the results indicated that there is no significant difference between the two series (0.142 > 0.05). Also, the ANOVA test was performed and F value was 1.27161 larger than 0.05, confirming that there is no significant difference between the averages of the two series.

These results confirmed, furthermore, the capacity of the polymeric platform to detect FA in complex biological or pharmaceutical matrices after the sample preparation steps (extraction, filtration, and dilution to the desired concentration in the second case).

The results obtained are in agreement with those mentioned in the literature, as it can be seen from Table 3. The reported configurations are based on glassy-carbon or carbon paste electrodes and different metallic modifiers and the detection is performed mainly by square wave voltammetry (SWV) or DPV. In the case of pharmaceutical samples, the recoveries ranged from 94% to 105%. The linear range is generally narrower and the LOD smaller when the quantification of FA was performed at an acidic pH, because the electrochemical reduction of the analyte is dependent on the pH [25]. Furthermore, this work presents the ability of our sensor for in situ detection, due to the use of screen-printed electrodes. The data generated when working close to the physiological value of the pH are similar with our work, confirming furthermore the capacity of the platform based on conductive polymeric structures to assess FA from pharmaceutical samples.

## 4. Conclusions

A polymeric platform based on a carboxylic conductive polymer was developed via an electrochemical method without the use of any templates. The generated sensor was able to quantify FA from biological and pharmaceutical samples on a wide concentration range (2.5–200 μM) with very good recoveries. The novelty of this work resides in the simplicity of the generation method and the long shelf-life when kept at an established temperature. Also, the simple testing protocol that involves a two-step procedure: pretreatment and detection step in the same testing sample makes it suitable for the development of a portable device. Furthermore, the electrochemical results of the sensor were confirmed with a spectrophotometric procedure. Thus, this sensor represents a first promising step towards the elaboration of a point-of-care device that can be used in the biomedical, pharmaceutical, and even food control fields for the rapid monitoring of FA.

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
