# Peer review of "Selective Detection of Folic Acid Using 3D Polymeric Structures of 3-Carboxylic Polypyrrole"

_sensors, 2020, doi:10.3390/s20082315_

Round 1
Reviewer 1 Report
1.It would be easier to understand if the equivalent circuit was inserted in the figures
2.From SEM, it can be seen that 3D structures (200 nm) dispersed sparsely on the already rough surface of the carbon-based screen printed electrode. How much does 3-carboxylic pyrrole cover graphite electrode?
3.The subtitle should be carefully arranged in order to display the results clearly. The titles of 3.1 and 3.1 are the same, so it should be a mistake. “3.1. Electrochemical Deposition and Characterization of the Polypyrrole Morphology; 3.3. Electochemical Deposition and Characterization of the Polypyrrole Morphology”
4.The working principle should be stated in an independent part. The effects of 3D Polypyrrole should be explained clearly as well as the detection strategy.
Author Response
Please find the Answer to Reviewer 1 attached.

Reviewer 2 Report
The manuscript “Selective detection of Folic Acid using 3D polymeric 3 Structures of 3 carboxylic Polypyrrole” presents interesting and valuable work, which is within the scope of the Journal.
According to the anti-plagiarism tool the plagiarism percentage for this manuscript is moderate (approximately 15%, 146 matches from 104 sources). This application flagged specific sentences in our text and provided reference information about the source, calculated an overall originality score. Several sources (scientific publications) are highlighted due to plagiarism with percentage of 15%. I recommend to the authors to rewrite some parts of manuscript.
The major deficiency of manuscript is last section as suitable comparison with other analytical techniques is missing. I suggest the authors to evaluate (expand) references and to use relevant and recent ones. In the literature, the methods involving the individual determination of folic acid with different electrodes were reported (eg. J. Electroanal. Chem. 808 (2018) 189–194.), also analytical method developed for simultaneous determination of determination of mesalazine and folic acid using chitosan coated carbon nanotubes functionalized with amino groups in pharmaceutical products and biological samples (Journal of Electroanalytical Chemistry 851 (2019) 113450) is published.
I am not a native speaker myself, but even with my not-native level of English it is clear to me, that the manuscript need professional corrections and proof-reading.
Reviewer 3 Report
The manuscript is of interest and corresponds to the Journal aims and scopes. Folic acid quantification is important for medicine and pharmaceuticals quality control. The approach applioed demonstrates applicability of electropolimerized materials for the detection of folic acid. The manuscript can be accepted to publication after revision.
1. English needs revision in style and grammar (the badly constructed phrases have to be rewritten).
2. There are a number of technical mistakes throughout the manuscript:
- Fig. 4 caption is incomplete (part f is lost);
- Table 3 is presented as Table 2 on P. 14;
- Duplication of the paragraph on P. 10, lines 339-345 and 346-352;
- Description of sample preparation on lines 419-426 of Section 3.7.2. and lines 126-130 of section 2. Remove this text from Section 3.7.2.
3. Fig. 2b, the fitted data for EIS in the form of the corresponding lines should be presented on the plot in order to see the accuracy of fitting. Moreover, data in Table 2 are presented as single measurement results that is unacceptable. The sorresponding SD for each parameter as well as error of fitting as χ2 parameter should be added to Table 2.
4. Section 3.3. heading seems incorrect. The folic acid behavior is descussed in it but not electrodeposition and surface morphology.
5. Fig. 4e, 4f, and 5b, each point should have ±SD. Otherwise, the single measurement result is shown that is unrealiable.
6. The comparison of the sensor developed with spectrophotometry is incomplete. In order to confirm the similar precion t- and F-tests should be performed and corresponding results added to the text.
